METHODS

# Inferring microbial interactions with their environment from genomic and metagenomic data

**James D. Brunner** [1,2]*, **Laverne A. Gallegos-Graves** [1], **Marie E. Kroeger** [1¤]

**1** Biosciences Division, Los Alamos National Laboratory, Los Alamos, New Mexico, United States of America, **2** Center for Nonlinear Studies, Los Alamos National Laboratory, Los Alamos, New Mexico, United States of America

¤ Current address: In-Pipe Technology, Wood Dale, Illinois, United States of America
* jdbrunner@lanl.gov

**Data Availability Statement:** The genomes used in this work have been made available on the NCBI GenBank with accession numbers listed in S1 Table. All code for the method, as well as genome-

## Abstract

Microbial communities assemble through a complex set of interactions between microbes and their environment, and the resulting metabolic impact on the host ecosystem can be profound. Microbial activity is known to impact human health, plant growth, water quality, and soil carbon storage which has lead to the development of many approaches and products meant to manipulate the microbiome. In order to understand, predict, and improve microbial community engineering, genome-scale modeling techniques have been developed to translate genomic data into inferred microbial dynamics. However, these techniques rely heavily on simulation to draw conclusions which may vary with unknown parameters or initial conditions, rather than more robust qualitative analysis. To better understand microbial community dynamics using genome-scale modeling, we provide a tool to investigate the network of interactions between microbes and environmental metabolites over time.

Using our previously developed algorithm for simulating microbial communities from genome-scale metabolic models (GSMs), we infer the set of microbe-metabolite interactions within a microbial community in a particular environment. Because these interactions depend on the available environmental metabolites, we refer to the networks that we infer as *metabolically contextualized*, and so name our tool MetConSIN: Metabolically Contextualized Species Interaction Networks.

## Author summary

We present a method for analysis of community dynamic flux balance analysis by constructing an interaction network between microbes and metabolites in a microbial community. To do so, we reformulate community wide dynamic flux balance analysis as a sequence of ordinary differential equations, which can in turn be interpreted as networks. We then provide the sequence of interaction networks which depend on and dynamically alter the available metabolite pool, as well as the time-averaged network over the course of simulated growth on a finite resource medium.

scale models for the 10 genomes, is available at
https://github.com/lanl/metconsin.

**Funding:** JDB, LAGG, and MEK were supported by
the U.S. Department of Energy Biological System
Science Division Science Focus Area Grant
2019SFAF255 (PI: MEK). The funders had no role
in study design, data collection and analysis,
decision to publish, or preparation of the
manuscript.

**Competing interests:** The authors have declared
that no competing interests exist.

This is a *PLOS Computational Biology* Methods paper.

## Introduction

Microorganisms have profound impacts on ecosystems ranging from the human gut to forest
soils to plant root systems. In humans, recent advances in technology have created a plethora
of works describing the differences in microbial community, composition, and function
between diseased patients and healthy controls [1–7], clearly demonstrating that microbial
communities play an important role in human health. Likewise, environmental microbial
communities have been found to affect biogeochemical cycling in soil [8], leading to changes
in plant decomposition and soil carbon sequestration that effect the amount of greenhouse
gases in the atmosphere. Even in plants, rhizosphere microbial communities affect growth and
resilience [9] as well as response to drought [10].

To understand and predict the effects of microbial communities on their environment, we
must first understand how these communities assemble and interact. Biotic interactions
between microbes are a driving force in community assembly, with both positive and negative
interactions between microorganisms creating different community compositions. Moreover,
the ability for non-resident microorganisms to invade the community is also largely controlled
by biotic interactions [11], which makes it critical to understand these interactions to accu-
rately predict treatment success for microbiome manipulation. It is well established that com-
munity structure is important in determining the impact of the microbiome on its host
environment. For example, disease-free asymptomatic individuals will have pathogenic bacte-
ria in their microbiome [7], suggesting that community structure and microbial interactions
affect the host-microbe relationship.

The advent of modern sequencing and metabolic pathway analysis has led to an effort to
organize this data into useful models of microbes and microbial communities. These models,
which represent mathematically the internal network of chemical reactions within a cell's
metabolism are called genome-scale metabolic models (GSMs) [12, 13]. GSMs and the con-
straint-based reconstruction and analysis (COBRA) methods that make use of them have
shown growing promise in predicting and explaining the structure and function of microbial
communities [14–17]. However, the complexity of these models means that analysis is often
based only on simulation, and is very sensitive to parameters and other assumptions. For
example, many modern community metabolic modeling methods seek to predict co-culture
growth or biomass at chemostatic equilibrium using artificial community-wide constraints
[18–20]. On the other hand, dynamic methods that use predictions about growth rate and
metabolite consumption to construct a dynamical system suffer from dependence on
unknown metabolic parameters and initial conditions, as well as heavy computational cost
[21–23]. In fact, most tools for community modeling only provide predictions of species
growth rates and metabolite consumption, without providing an understanding of the funda-
mental interactions that lead to these predictions [24, 25]. Some qualitative insight into the sys-
tems is possible using simulated knock-out experiments [20, 26] or simplifying the system
[27]. However, new methods for qualitative analysis of community metabolic models are
needed.

An interaction network provides an interpretable object that can be used to characterize a
microbial community in more depth than composition alone [28], and suggest keystone taxa
and other functional properties of the community [29, 30]. These advantages, and the apparent

importance of microbial interactions, have led to the use of network inference and analysis for understanding important phenomena including disease treatment [31] and human impact on the climate [32]. The most commonly used method for network inference involves computing the propensity for microbes to appear together in a sample, most commonly defined by co-occurrence frequency, correlation, or covariance [33, 34]. More sophisticated methods for inferring associations between microbes include the use of regression-based and probabilistic models [28, 35] or fitting to time-longitudinal data [36]. Additionally, some modern methods have sought to combine mechanistic hypotheses with statistical network building using machine learning approaches by incorporating "background knowledge" of known microbial interactions [37] or using simple microbial characteristics along with a set of known interactions [38].

GSMs and COBRA modeling provide an attractive avenue for a "bottom up" approach to network building from underlying metabolic mechanism [39]. This can be done using simulated knock-out experiments [20], but this approach suffers from a focus on direct microbe-microbe interactions, which lead to models that lack the complexity of full metabolite mediated networks [40, 41]. While differences in networks across meta-groups may provide insight, these networks in general provide few avenues for prediction and design. Patterns in network structure cannot be directly related to function without further study, and networks built in this way cannot account for dynamically changing interactions across perturbations in the environment.

In this manuscript, we present a method for inferring interactions between microbes and metabolites within a microbial community by leveraging genome-scale metabolic models (GSMs). This method requires only some method of constructing GSMs as well as an estimate of the metabolic environment of the community. GSMs can be built as long as a genome can be assigned to each member of the community, using automated construction methods such as *CarveME* [42] or *modelSEED* [43]. Assigning genomes to community members in a sample can be done with genomic or metagenomic data, or if that data is not available, a less accurate assessment can be done by matching amplicon sequence data with previously characterized genomes.

Our method is based on *Flux balance analysis (FBA)*, which allows us to infer microbial growth and exchange of metabolites with the environment. These can be combined into a dynamical system, called *dynamic flux balance analysis (DFBA)* which in turn can be represented as a sequence of networks. Simulation of dynamic flux balance analysis requires the solution to a linear optimization problem at each time-step. These solutions can be found without repeated optimization by using a *basis* for an initial solution, which allows us to find new solutions as the problem constraints change simply by solving a linear system of equations. This means that we can reformulate the dynamical system as an *ordinary differential equation (ODE)* that has solutions that match the solution to the DFBA problem for some time interval. Finally, this ODE system can be naturally interpreted as a network of interactions between microbes and metabolites, achieving our goal. We note also that this ODE system provides a second network of interactions between the metabolites that is mediated by the microbial metabolisms of the community. Fig 1 provides a graphical summary of the method.

## Background

### Dynamic flux balance analysis

Advances in genetic sequencing have led to the construction of genome scale models (GSMs) of the metabolic pathways of microbial cells, and to methods to analyze and draw insight from such large scale models [12]. Constraint based reconstruction and analysis (COBRA) is used to

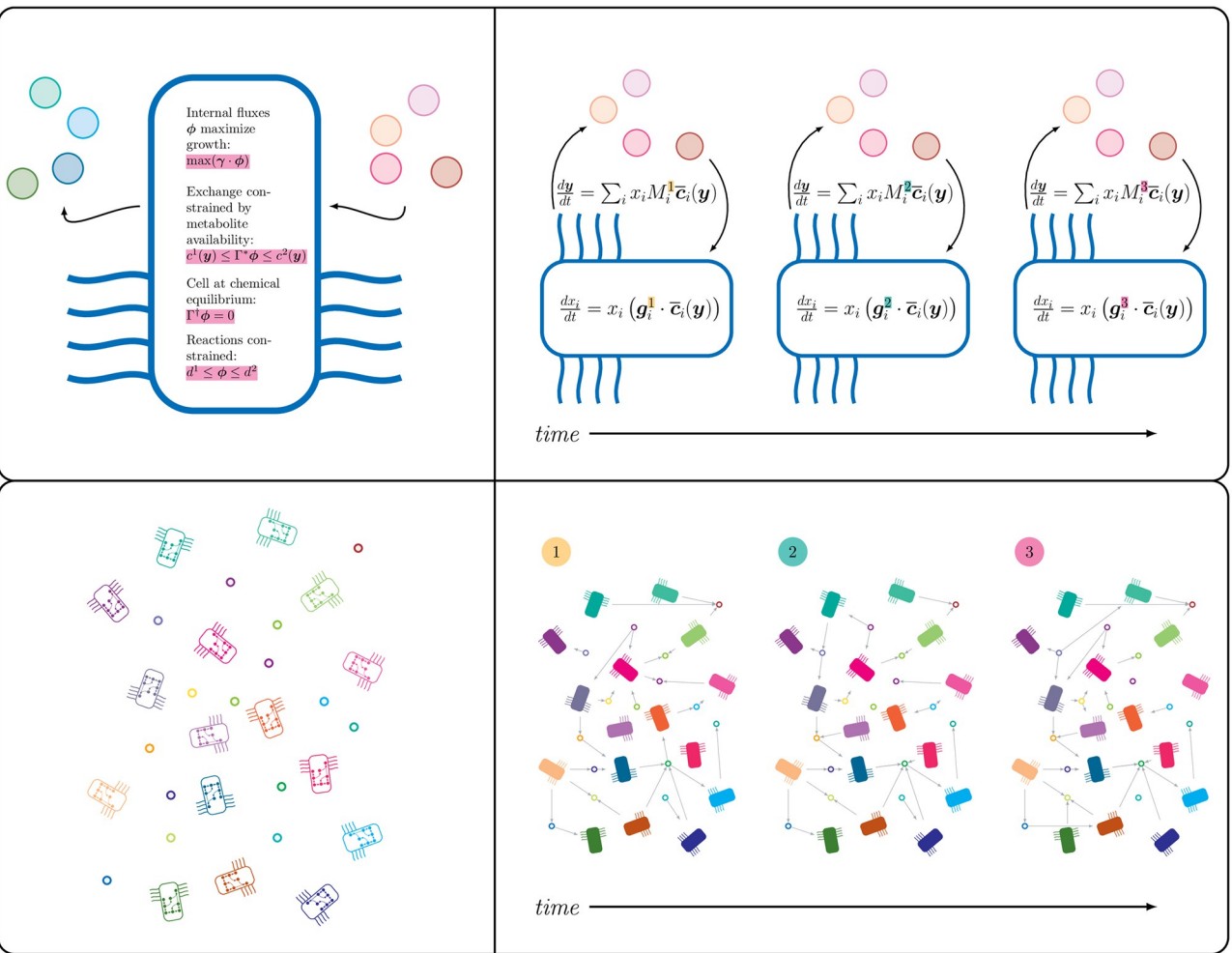

**Fig 1.** MetConSIN reformulates dynamic flux balance analysis (represented by the top-left figure) as a series of smooth ordinary differential equations (top-right). This dynamical system provides an model of a microbial community and its environment (represented by the bottom-left figure) as a series of metabolite mediated networks (bottom-right).

model steady state fluxes $\psi_i$ through a microorganism's internal metabolic reactions under physically relevant constraints [12]. *Flux balance analysis* (FBA) is a COBRA method that optimizes some combination of internal reaction fluxes which correspond to increased cellular biomass, subject to the constraint that the cell's internal metabolism is at equilibrium.

Precisely, flux balance analysis assumes that cell growth and metabolic flux can be determined by solving the following linear program [23]:

$$
\left\{
\begin{aligned}
\max(\boldsymbol{\psi} \cdot \boldsymbol{\gamma}) \\
\Gamma^{\dagger}\boldsymbol{\psi} = 0 \\
\boldsymbol{c}^1(\boldsymbol{y}) \leq \Gamma^{*}\boldsymbol{\psi} \leq \boldsymbol{c}^2(\boldsymbol{y}) \\
\boldsymbol{d}^1 \leq \boldsymbol{\psi} \leq \boldsymbol{d}^2
\end{aligned}
\right\}
\tag{1}
$$

where the matrices $\Gamma^*$, $\Gamma^{\dagger}$ together represent the stoichiometry of the cell's metabolism, the vector $\boldsymbol{\psi}$ represents the flux through the cell's internal reactions, the objective vector $\boldsymbol{\gamma}$ encodes

the cell's objective, exchange constraints $c^1$, $c^2$ are determined in part by available external metabolites and internal constraints $d^1$, $d^2$ are known. Exchange rates $v_j$ of metabolite $j$ between the cell and its environment are in turn determined by internal flux according to $v = \Gamma^* \psi$. For convenience, we define a vector

$$c = (c_1, d_1, c_2, d_2, 0) \tag{2}$$

to be the vector of all of the problems constraints.

Solutions to FBA provide a rate of increase of biomass which can be interpreted as a growth rate for a cell. Furthermore, FBA solutions allow us to compute the vector $v$, which represents metabolite exchange between the cell and an external metabolite pool. By assuming that constraints on nutrient exchange reactions within the metabolic network are functions of the available external metabolites, the coupled system of microbe and environment can be modeled. For a community of microbes $x = (x_1, \ldots, x_p)$ in an environment defined by the concentration of nutrients $y = (y_1, \ldots, y_m)$ this model has the form [23]:

$$\frac{dx_i}{dt} = x_i(\gamma_i \cdot \psi_i) \tag{3}$$

$$\frac{dy_j}{dt} = -\sum_{i=1}^{p} x_i(\Gamma_i^* \psi_i)_j \tag{4}$$

with $\psi_i$ determined separately for each organism according to a linear program of the form Eq (1). This system is referred to as *dynamic flux balance analysis (DFBA)*. Note that this is a metabolite mediated model of the community, meaning that the coupling of the growth of the separate microbes is due to the shared pool of metabolites $y$.

## Piece-wise smooth representation

Simulation of the dynamical system given by Eqs (3) and (4) can be accomplished by leveraging the *fundamental theorem of linear programming* [23], which states that if Eq (1) has an optimal solution, then it has an optimal solution that can be represented as the solution to an invertible system of linear equations [44]. This means that there is some invertible matrix $B$ and index set $\mathcal{B}$ such that

$$\psi = B^{-1}\bar{c} \tag{5}$$

is an optimal solution to the linear program (where for ease of notation we substitute $\bar{c} = c_{\mathcal{B}}$). The key observation allowing efficient forward simulation of Eqs (3) and (4) is that as the constraints $c^1(y)$, $c^2(y)$ vary, the matrix $B$ does not change. In other words, there is some time interval such that we can replace the linear program Eq (1) with the linear system of equations

$$B_i \psi_i = \bar{c}_i(y) \tag{6}$$

for some time-interval, where $\bar{c}_i$ is a subset of the bound functions $c_i$. At the end of this time interval, the solution to Eq (6) stops obeying the problem constraints, and new $B_i$ must be chosen. Putting this together, we can define a sequence of time intervals $[t_0, t_1)$, $[t_1, t_2)$, $\ldots$, $[t_{n-1}, T)$ such that solutions to the system defined by dynamic FBA for a community (Eqs (1), (3)

and (4) are solutions to the system of ODEs

$$\frac{dx_i}{dt} = x_i(\boldsymbol{\gamma}_i \cdot (B_i^k)^{-1}\bar{\boldsymbol{c}}_i^k(\boldsymbol{y})) \tag{7}$$

$$\frac{d\boldsymbol{y}}{dt} = -\sum_{i=1}^{p} x_i \Gamma_i^* (B_i^k)^{-1}\bar{\boldsymbol{c}}_i^k(\boldsymbol{y}). \tag{8}$$

on the interval $[t_k, t_{k+1})$ for some invertible matrices $B_i^k$.

The challenge of efficient forward simulation of DFBA is then in finding the matrices $B_i^k$, which may be non-unique. In previous work, we presented a method for choosing the set of $B_i^k$ that allow forward simulation so that $t_{k+1} > t_k$, and created a python packaged called *SurfinFBA* for simulation. In brief, it is necessary to solve a new optimization problem defined by the time-derivative of the constraints of the original FBA linear program whenever a new $B_i^k$ is needed. Furthermore, we have recently improved this method to increase the length of the time intervals $[t_k, t_{k+1})$ (see supporting material S1 Text). This improvement is packaged with the MetConSIN package, which includes SurfinFBA.

## Methods

### MetConSIN network construction

The dynamical system defined by DFBA (Eqs (1), (3) and (4)) can be simulated but is difficult to interpret and analyze, especially when accounting for uncertainty in initial conditions and bound functions $\boldsymbol{c}_i^1(\boldsymbol{y}), \boldsymbol{c}_i^2(\boldsymbol{y})$. However, Eqs (7) and (8) suggest that the system can be interpreted as a network of interactions between microbes and metabolites on the time interval $[t_k, t_{k+1})$ with only mild assumptions on $\boldsymbol{c}_i^1(\boldsymbol{y}), \boldsymbol{c}_i^2(\boldsymbol{y})$. DFBA therefore implies a sequence of interaction networks representing the dynamics of a microbial community. Furthermore, the FBA solutions for each community member, and as a result the interactions that can be inferred, depend entirely on the metabolic environment ($\boldsymbol{y}$). Therefore, for a fixed metabolic environment (created by, e.g., flowing metabolites into a bioreactor at an increasing rate or finding a chemostatic community equilibrium) DFBA provides an interaction network representing community metabolic activity.

Without loss of generality, we may construct Eq (1) so that the forward direction (i.e. positive flux) of each of the first $m$ reactions transports one of the $m$ environmental metabolites into the cell. Then we assume that $\boldsymbol{c}^2(\boldsymbol{y})$ has the form

$$\boldsymbol{c}_i^2(\boldsymbol{y}) = (c_{i1}^2(y_1), \ldots, c_{im}^2(y_m)) \tag{9}$$

with non-decreasing $c_{ij}^2(y_j)$, and that $\boldsymbol{c}_i^1(\boldsymbol{y}) = \boldsymbol{c}_i^1$ is constant. In plain language, we assume that the fluxes of reactions that transport environmental metabolites into the cell are bounded by the availability of the corresponding environmental metabolites, and the other reactions have constant bounds.

Under this assumption, for time-interval $k$, Eq (7) can be rearranged into the form

$$\frac{dx_i}{dt} = C_i^k x_i + \sum_{j=1}^{m} a_{ij}^k x_i c_{ij}^2(y_j) = x_i\left(C_i^k + \sum_{j=1}^{m} a_{ij}^k c_{ij}^2(y_j)\right) \tag{10}$$

where $C_i^k$ is a constant that we refer to as *intrinsic growth*, and the $a_{ij}^k$ are combinations of

entries in $\gamma_i$ and $(B_i^k)^{-1}$. Likewise, Eq (8) can be rearranged into the form

$$\frac{dy_l}{dt} = -\sum_{i=1}^{p}\left(D_{il}^k x_i + \sum_{j=1}^{m} b_{ijl}^k x_i c_{ij}^2(y_j)\right) \tag{11}$$

where $D_{il}^k$ is a constant, and the $b_{ijl}^k$ are entries of the matrix $\Gamma_i^*(B_i^k)^{-1}$. We may now interpret these ODEs as networks of interactions term by term.

 Eq (10) can be interpreted as growth of a microbial population proportional to the population biomass, with growth rate modified by the environmental metabolites $y_j$. This is similar to a generalized Lotka-Volterra model [41], and can be naturally represented by a set of network edges pointing from a metabolite to the microbe with weights $a_{ij}^k$.

 The terms in Eq (11) are slightly more complicated to interpret. The terms $D_{il}^k x_i$ represent some effect of the microbe $i$ on the available biomass of $j$ over the time interval $[t_k, t_{k+1})$ which only changes with the biomass $x_i$ of microbe $i$ over this interval. This effect is the result of growth pathways that do not depend on metabolite availability, and may be 0. Additionally, when $j = l$, the term $b_{ill}^k x_i c_{il}^2(y_l)$ can be interpreted as pairwise interactions between microbe $i$ and metabolite $l$, e.g. consumption of a carbon source. These two sets of terms can be represented by a set of network edges pointing from the microbe to the metabolite, with weights $D_{il}^k + b_{ill}^k$. The remaining terms represent interactions that involve two metabolites and one microbe. In the formalism of interaction network theory (see [45–48]), these remaining terms can be interpreted as reactions of the form

$$X_i + Y_j \xrightarrow{b_{ijl}^k} X_i + Y_j + Y_l \tag{12}$$

if $b_{ijl} < 0$ (and so the interaction increases the available biomass of metabolite $l$), meaning that microbe $i$ and metabolite $j$ interact to form metabolite $l$. This means, e.g., that metabolite $l$ is created as a byproduct of the metabolism of metabolite $j$ by microbe $i$. In this case, we again represent the interaction as a network edge from microbe $i$ to metabolite $l$, but now annotate the edge with the information that this interaction is mediated by metabolite $j$. Finally, we may do the same if $b_{ij} > 0$, although we note now that this represents a non-autocatylitic interaction, meaning that the available biomass of metabolite $l$ is reduced independent of the current available biomass. While this seems counter-intuitive, it arises when metabolite $l$ is consumed in some metabolic pathway but it is not the rate-limiting external metabolite for that pathway. In fact, when enough of the biomass is consumed so that metabolite $l$ becomes rate-limiting, the system will transition to the next interval $[t_{k+1}, t_{k+2})$ and the ODEs Eqs (10) and (11) will change. This transition ensures that the non-negative orthant is forward invariant for the DFBA system, meaning the system will not achieve a non-physical state of negative biomass. The mapping from the ODEs to a microbe-metabolite network is summarized in Table 1.

## Metabolite interaction network construction

In addition to the microbe-metabolite interaction network described above, the last set of interactions suggest a second network can be formed which includes only the metabolites. In the microbe-metabolite network, the terms $b_{ijl}^k x_i c_{ij}^2(y_j)$ in Eq (11) describe how microbe $i$ effects the available biomass of metabolite $l$ as mediated by metabolite $j$. We may instead interpret this as the metabolite $j$ effecting the available biomass of metabolite $l$ through reactions carried

**Table 1. Mapping of terms in Eqs ([10]) and ([11]) to network edges.**

| Equation | Term | Network Edge | Weight | Mediated by |
|---|---|---|---|---|
| $\frac{dx_i}{dt}$ | $a_{ij}x_i c_{ij}^2(y_j)$ | $Y_i \to X_i$ | $a_{ij}^k$ | - |
| $\frac{dy_l}{dt}$ | $D_{il}^k x_i$ | $X_i \to Y_l$ | $D_{il}^k$ | - |
| $\frac{dy_l}{dt}$ | $b_{ill}^k x_i c_{il}^2(y_l)$ | $X_i \to Y_l$ | $b_{ill}^k$ | - |
| $\frac{dy_l}{dt}$ | $b_{ijl}^k x_i c_{ij}^2(y_j)$ | $X_i \to Y_l$ | $b_{ijj}^k$ | $Y_j$ |

out by microbe $i$. This interpretation suggests a network of edges

$$Y_j \xrightarrow{\ X_i\ } Y_l$$

labeled by the microbe whose metabolism contributes the edge.

### Microbe interaction network construction

While dynamic FBA can be written as a series of microbe-metabolite interaction networks, researchers are often interested in the emergent interactions between microbes themselves. MetConSIN provides a simple heuristic for inferring these interactions based on the competitive and cross-feeding interactions of the microbe-metabolite network. The heuristic is as follows: to determine the effect of microbe $X_i$ on the growth of microbe $X_j$, we find the set of all paths of length two of the form

$$X_i \xrightarrow{\ w_{il}\ } Y_l \xrightarrow{\ w_{lj}\ } X_j$$

with weights in the microbe-metabolite network $w_{il}$ and $w_{lj}$. If $w_{il} < 0$, meaning that $X_i$ consumes or otherwise depletes $Y_l$, while $w_{lj} > 0$, e.g. $Y_l$ is a limiting resource for the growth of $X_j$, then this pair of interactions can be interpreted similarly to competition between $X_i$ and $X_j$, although this competition does not need to be symmetric. Conversely, if $w_{il} > 0$ and $w_{lj} > 0$, then the presence of $X_i$ will increase the growth of $X_j$ through cross-feeding. We therefore take as the composite edge weight $\tilde{w}_{ij}$ of the emergent interaction

$$X_i \xrightarrow{\ \tilde{w}_{ij}\ } X_j$$

to be the sum over all such paths of the products of the weights of the edges in the path:

$$\tilde{w}_{ij} = \sum_{l:X_i \longrightarrow Y_l \longrightarrow X_j} w_{il} w_{lj}. \tag{13}$$

### Sequencing of soil isolates

**Bacterial isolation.** Ten bacterial isolates were originally isolated using serial dilution from soils collected in Utah [49] (38.67485 N, 109.4163 W, 1310 elevation), New Mexico (35.4255167 N, 106.6498 W, 5405 elevation), and Colorado (37.23081667 N, 107.8599667 W, 6484 elevation). These bacterial isolates were grown on either Caulobacter medium, 1/10 Tryptic soy Medium (TSB), or Nutrient medium at 30oC for 24–72 hours depending on the strain. Single colonies were then transferred in their respective growth medium and grown for 24–72 hours at 30oC while shaking at 250rpm. Bacterial biomass was harvested from overnight cultures by centrifugation and High Molecular Weight (HMW) DNA extractions were completed using the Qiagen MagAttract HMW DNA Kit (Qiagen,

Hilden, Germany) following manufacturer's protocol. Two of the bacterial isolates required an additional clean-up after DNA extraction which was completed using the Qiagen PowerClean Pro Clean-up Kit (Qiagen, Hilden, Germany) following manufacturer's protocol.

**Library preparation.**    DNA library preparation and sequencing was completed at the LANL Genomics Facility as described in detail below. DNA initial quantification was done using Qubit High sensitivity ds DNA kit (Invitrogen). DNA integrity was assessed on Tape Station using gDNA Screen tape (Agilent Technology). DNA purity ratios were determined on NanoDrop 1 spectrophotometer (ThermoScientific). 1ug of genomic DNA for each sample was sheared using g-Tubes (Covaris, USA). Shearing parameters were chosen according to the DNA integrity of a particular isolate. All but two samples were sheared the following way: shear 2 min at 7,000 rpm, flip the tube, and shear 2 min at 7,000 rpm. More fragmented samples were sheared using the following parameters: shear 2 min at 3,500 rpm, flip the tube, and shear 2 min at 3,500 rpm.Sheared DNA was collected and purified using AMPure PB beads (Pacific Bioscience, USA) as per PacBio protocol. The quality and quantity of the purified DNA were assessed using the TapeStation and Qubit as described above. SMRT bell templates were constructed according to the PacBio protocol using Express Template Prep Kit 2.0.

First, DNA underwent damage repair, end repair and A-tailing. It was followed by the barcoded overhang adapter ligation and purification with 0.45X volume of AMPure PB beads (Pacific Bioscience, USA). The barcoded samples were pooled in the equimolar amount according to the volumes provided in the PacBio Microbial Multiplexing Calculator. The pooled SMRT bell library was quantified using the Qubit DNA HS kit (Invitrogen) and the average fragment size was determined on Bioanalyzer using the DNA HS kit (Agilent). The conditioned sequencing primer v. 4 was annealed and Sequel II DNA Polymerase 2.0 was bound to the SMRT bell library. The template/ DNA polymerase complex was diluted and purified with 1.2X volume of AMPure PB beads (Pacific Bioscience, USA). The complex was sequenced on PacBio Sequel II instrument, using 1 SMRT cell 8M and Sequencing chemistry 2.0, 30 hour movies were recorded.

**Sequencing.**    The raw PacBio reads were converted to PacBio HiFi reads using the "CCS with Demultiplexing" option in SMRTLink 11.0.0.146107. This resulted in a total of 1,532,731 HiFi reads for a total yield of 8.3 Gbp. The median read quality was Q37 with a mean read length of 5,409 bp.

The reads were assembled using Flye v.2.9-b1768. Putative number of plasmids were estimated by looking at the `assembly_info.txt` files output by Flye. This file indicates if a contig is circular and/or a repeat. Contigs that were indicated as circular but not a repeat, as well as under 500 Kbp were assumed to be plasmids. Contigs that had the same attributes but were over 500 Kbp were assumed to be a complete bacterial chromosome. Then, the assemblies were annotated using Prokka v.1.14.6. The taxonomy of the genomes were derived using gtdbtk v.1.5.0.

**Genome-scale model reconstruction & MetConSIN simulation.**    Genome-scale models for the 10 bacterial genomes were created using modelSEED [43] within the KBase computational platform [50]. The models were gap-filled with a complete media. The resulting models were used to test the MetConSIN simulation method, with models labeled according to the genome ID of the corresponding bacterial genomes. For the clarity of the network figures, we label the nodes corresponding to each model with the unique 1- or 2-digit integer that appears in the genome ID. Table 2 lists the IDs, classification, and node labels for the 10 models, and the supplemental file S1 Table contains details of the sequencing results.

**Table 2. The 10 bacterial genomes that were used to demonstrate the method were isolated from soil samples and sequenced with PacBio.** Genomes were assembled using the procedure detailed in the *methods* section and are available on the sequence read archive database. We constructed GSMs for these genomes using modelSEED, which are available in the *Examples* directory of the project repository. Full details of the sequencing results can be found in S1 Table.

| Barcode ID | NCBI Classification | NCBI Taxonomy ID | Node Label |
|---|---|---|---|
| bc1001 | Kocuria sp. ALI-2-A | 3025731 | 1 |
| bc1002 | Nocardioides sp. ALI-37-C | 3025730 | 2 |
| bc1003 | Williamsia muralis ALI-73-A | 85044 | 3 |
| bc1008 | Priestia megaterium s92 | 1404 | 8 |
| bc1009 | Paenibacillus spp. s49 | 3051831 | 9 |
| bc1010 | Microbacterium spp. s49 | 3025735 | 10 |
| bc1011 | Streptomyces sp. ALI-76-A | 3025736 | 11 |
| bc1012 | Mesorhizobium spp. s92 | 3051830 | 12 |
| bc1015 | Paenibacillus_E spp. s92 | 3051829 | 15 |
| bc1016 | Priestia megaterium strain s92 | 1404 | 16 |

## Results & discussion

### Simulation of 10 soil-isolated taxa

MetConSIN provides analysis of the dynamic flux balance analysis (DFBA) system by inferring a set of interaction networks from that system. To demonstrate this utility, we simulated the growth of 10 taxa isolated from soil using DFBA, and used MetConSIN to construct the series of interaction networks that the community behaved according to over the course of the simulation. The interaction networks simulated by MetConSIN were used to develop targeted hypotheses about microbial interactions that are currently being tested in the laboratory.

Fig 2 shows the simulated growth of genome-scale models of all 10 taxa in the simulated community on a finite media in an aerobic environment, all of which reached stationary phase when glucose was depleted. The community grew through a set of 3 distinct time-intervals, each with a corresponding species-metabolite and metabolite-metabolite network. These networks, as well as the time-weighted variance between them are shown in Figs 3 and 4. The two sets of networks provide a mechanistic explanation of the microbial growth and metabolic activity of the community. These networks tell us which microbes are consuming and producing environmental metabolites, as well as how the environmental metabolites effect cell growth. For any time interval, an edge from a metabolite to a microbe has a non-zero edge weight if and only if the simulated growth rate of the microbe is a function of the concentration of the metabolite during that interval. Inspection of the network reveals that only a few such edges exist, even though many metabolites are depleted by microbes. This is because only a subset of the constraints of flux balance analysis determine the growth rate, as indicated by the basic index set $\mathcal{B}$ that is used to solve DFBA. In other words, only rate-limiting metabolites appear as source nodes in the network.

Inspecting the network can reveal interesting time-dependent interactions. For example, we notice in Fig 4 that for $t \in [0.24, 0.42)$, community metabolism of D-Glucose causes consumption of Fumarate and production of Succinate. This interaction is very strong during this interval, which lies between two time-points at which the model of genome *bc1012* changes its network connections, so we might guess that this interaction is mediated by that model. MetConSIN provides edge data for each edge in the network, including in the case of the metabolite-metabolite networks which microbe mediated the interaction. Inspection of this output reveals that, indeed, the model for genome *bc1012* mediates the interactions between D-Glucose, Fumarate, and Succinate.

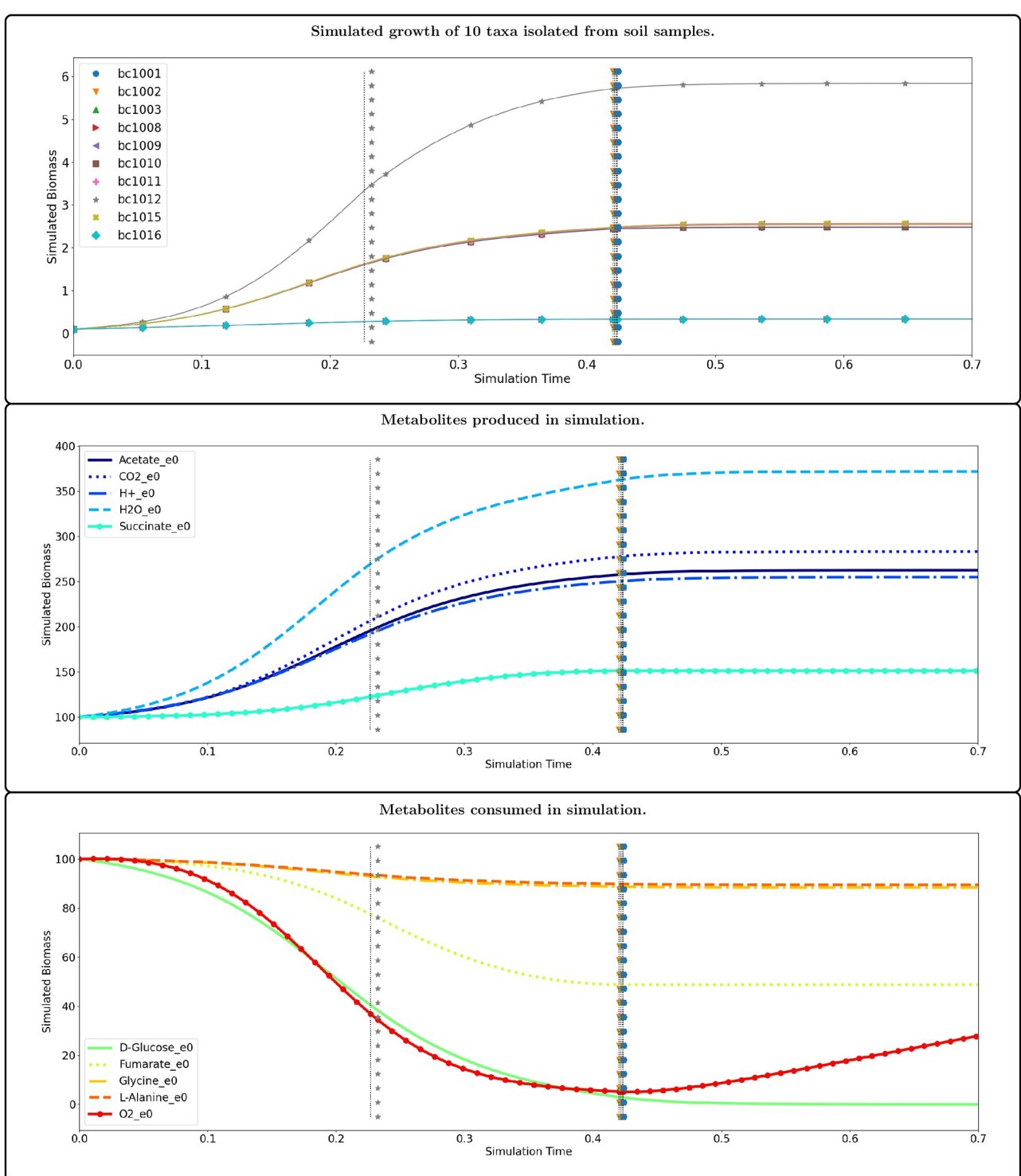

**Fig 2. The 10 taxa isolated from soil samples all grow to stationary phase at varying rates.** The community grows in 3 distinct time-intervals, each with a specific interaction network associated. The dotted lines indicate time-points at which SurfinFBA required a new basis for forward simulation of at least one taxa. When only one new basis was needed, the color of the dotted line corresponds to which taxa required a new basis. A black dotted line (e.g. at $t = 0.226$) indicates that no new basis was computed, but step size needed to be reduced. Only ten metabolites varied in simulated biomass by more than 1%; five were produced by the simulated community and five were consumed. The remaining environmental metabolites did not vary in simulated biomass by more than 1%. All metabolites were initially set to the same value and an aerobic environment was simulated by constant inflow of oxygen. See the *Example* folder of the project repository [51] for exact simulated conditions and full simulated results.

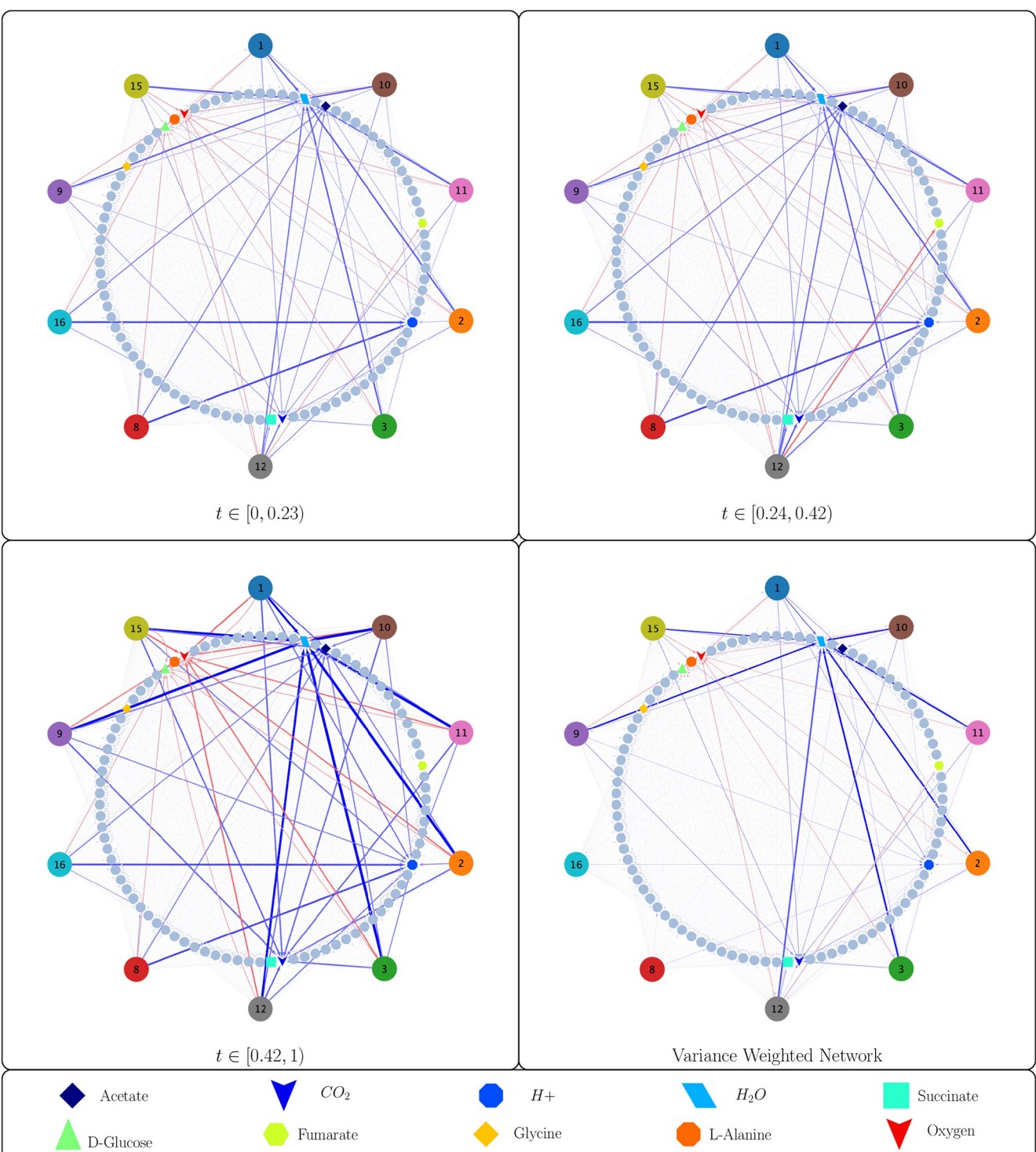

**Fig 3. A community of 10 taxa (detailed in Table 2) from soil and modeled using ModelSEED behaves according to a series of 3 interaction networks.** One powerful application of MetConSIN is an inspection of how the community changes its qualitative behavior as time proceeds. To illustrate the differences, we show here the networks corresponding to each significant time interval as well as a network with edges weighted by the variance of the edges across those intervals. In the three networks corresponding to time intervals, edge color represents the sign of the interaction, with red representing negative and blue representing positive, while edge thickness represents interaction strength. In the variance-weighted network, edge color represents the sign of the time-averaged interaction, while edge thickness represents variance of that interaction across the time intervals. The 10 microbial taxa and the metabolites that are significantly produced or consumed are colored to match Fig 2, and the remaining environmental metabolites are shown with as partially transparent.

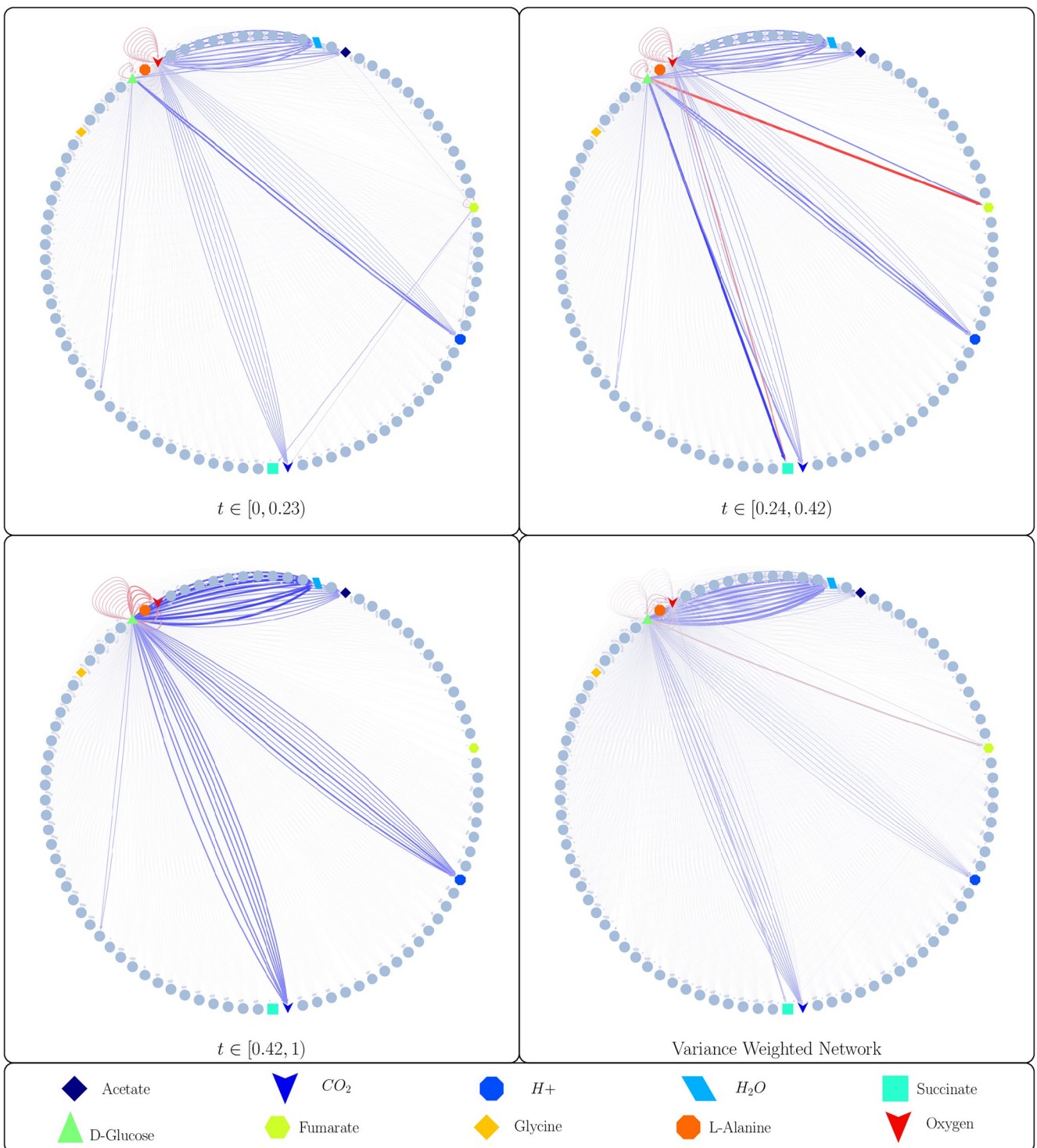

**Fig 4. MetConSIN can infer the metabolic activity of the 10 member community (detailed in Table 2) as represented by the emergent interactions between metabolites due to microbial metabolisms.** This network is not constant over time, and changes with the species-metabolite networks. These networks make clear the importance of D-Glucose (green triangle) and Oxygen (red arrowhead) as rate-limiting metabolites in the community's metabolic activity. Here, we show the network for each significant time-interval, as well as a network with edges weighted by the variance of the edges across those intervals. In the three networks corresponding to time intervals, edge color represents the sign of the interaction, with red representing negative and blue representing positive, while edge thickness represents interaction strength. In the variance-weighted network, edge color represents the sign of the time-averaged interaction, while edge thickness represents variance of that interaction across the time intervals.

The two major transitions in the simulation both involved a series of basis-changes, meaning that one or more microbes altered their connectivity in the network, and MetConSIN can provide details as to why and how these transitions happened. For example, in the first transition, in which the model of genome *bc1012* altered its connectivity, MetConSIN reports that the first transition happened because the solution violated the upper bound of glucose uptake. Prior to this transition, glucose was abundant enough that it did not limit *bc1012* growth. After this transition, the constraint on glucose uptake was activated by the model along with the reaction for lactate oxidation. The second transition occurred immediately after the first, when the reaction dehydrogenating NADH violated its lower bound, causing deactivation of this reaction and the fumarate exchange upper bound constraint. The result of these internal changes can be observed in the interaction networks simply by inspecting the difference in the network edge weights (where we may assign the weight 0 to an edge that is not present in a network). The largest (in magnitude) changes were in *bc1012*'s production of succinate and consumption of fumarate, both of which were reduced by about 77%. We display which microbe changed its network connectivity at each transition with the style and color of the dashed lines in Fig 2 that indicate the time-points at which the transitions occurred.

MetConSIN's analysis provides an avenue for using dynamic FBA to infer how microbes interact and how these interactions vary with community composition and over time. For example, we can infer from MetConSIN that the ten taxa whose genomes we isolated from soil behave antagonistically due to competition for resources, as seen in Fig 5(a) and 5(b).

MetConSIN's microbe-microbe interactions are based on a simple heuristic meant to identify competition and cross-feeding. This works well if an interaction between two microbes is based on a single metabolite, but simply summing the interactions is likely not the best approximation. In future work, we plan to define a more rigorous simplification of the metabolite-mediated system as a direct microbe interaction system and characterize the error of this simplification.

Our ten-member community showed only negative interactions in part because the genome-scale models that we used include the core metabolism of each taxa, making competition easy to identify, but do not include many details on the production of secondary

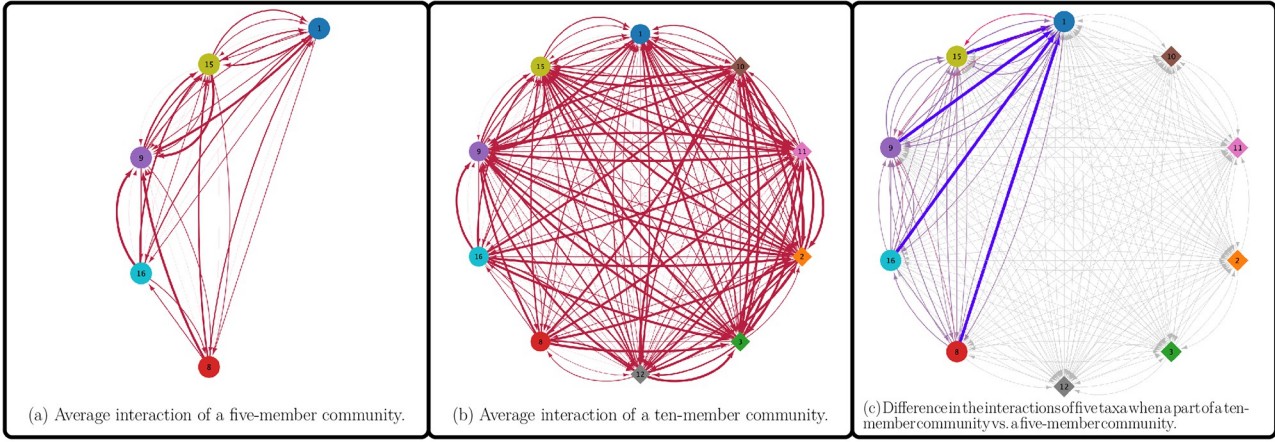

(a) Average interaction of a five-member community.   (b) Average interaction of a ten-member community.   (c) Difference in the interactions of five taxa when a part of a ten-member community vs. a five-member community.

**Fig 5.** The models of the ten genomes isolated from soil behave antagonistically. Figure (a) shows the time-averaged antagonistic interactions of a subset of five models, while figure (b) shows the time-averaged antagonistic interactions of the ten models simulated together. Figure (c) shows the difference in the time-averaged interactions common to the two networks, with line width corresponding to absolute difference, and color corresponding to signed-difference, where bluer shades represent a stronger interaction in the five model simulation and redder shades represent a stronger interaction in the ten model simulation.

metabolites. Secondary metabolites are compounds produced by bacteria that do not have a direct role in cell growth, but can have a profound impact on community organization [52–54]. Genome-scale modeling often focuses on the core metabolism and growth of an organism, meaning that these metabolites are often missing. This omission is a major challenge for any method that seeks to use GSMs to study microbial ecology. For MetConSIN to incorporate interactions mediated by secondary metabolites, the GSMs used must already include pathways that produce these metabolites. Furthermore, FBA constraints must be carefully chosen so that models do not simply ignore secondary metabolites in favor of immediate growth. As genome-scale models improve to include secondary metabolite production, MetConSIN can likewise be improved to infer interactions from secondary metabolites.

We observe antagonistic interactions in all of the subsets of the community that we simulated in isolation, but the strength of the competition may vary with different community composition. Indeed, Fig 5(c) shows that the implied relationships emerging from competition for resources are not the same in a five-model subset of the community as when these five models are simulated as part of the larger community of 10 models.

The species-metabolite and metabolite-metabolite networks provided by MetConSIN offer mechanistic insight into the metabolic activity of microbial communities, including identification of how metabolic connections change with community composition. In Fig 6, we investigate the strengths of the various connections one model, *bc1001*, had in networks produced by MetConSIN for various communities involving *bc1001*. For example, when grown in simulated coculture with *bc1016* and *bc1009*, *bc1001* tended to form weaker network connections than when grown in other combinations. Interestingly, when *bc1001* was grown in simulated coculture with *bc1015* and *bc1009*, it formed stronger connections compared to when simulated with *bc1016* and *bc1009*, even though switching *bc1015* for *bc1016* had little effect in other combinations. These connection differences are a possible mechanistic explanation for differential metabolic activity between communities, and suggest which community combinations should be prioritized in experimental design. For example, the results discussed above and displayed in Fig 6 suggest that *bc1001*, *bc1015* and *bc1009* undergo some kind of three-way interaction. Growth experiments with *bc1001*, *bc1015* and *bc1009* may therefore yield interesting results, especially if metabolomic data is collected in order to identify the three-way interactions taking place.

## Simulation of growth experiments & empirically observed interactions

In order to assess the accuracy of the DFBA simulation underlying MetConSIN and the networks produced, we simulated a community of organisms study in Weiss et al. [55]. In that work, the authors performed paired growth experiments in order to infer a network of interactions between 12 microbial taxa found in the Oligo-Mouse-Microbiota [56, 57] by comparing paired growth to lone growth on the same media. Furthermore, the authors used metabolomics data from their growth experiments to construct genome-scale metabolic models for these 12 taxa.

We tested our method by predicting the results of the paired growth experiments using DFBA simulation and using MetConSIN to construct metabolic-based interaction networks from pairwise simulations. We then compared MetConSIN's results to the growth data and log-ratio of pair and monoculture growth for each organism, which Weiss et al. use to infer an interaction network. These experiments revealed that DFBA has limited predictive power without model refinement, highlighting the need for interpretation to reveal the shortcomings of the underlying GSMs. DFBA predictions of the relative abundance of pair growth was often backwards, in the sense that DFBA predicted the lower-abundance microbe to be higher-

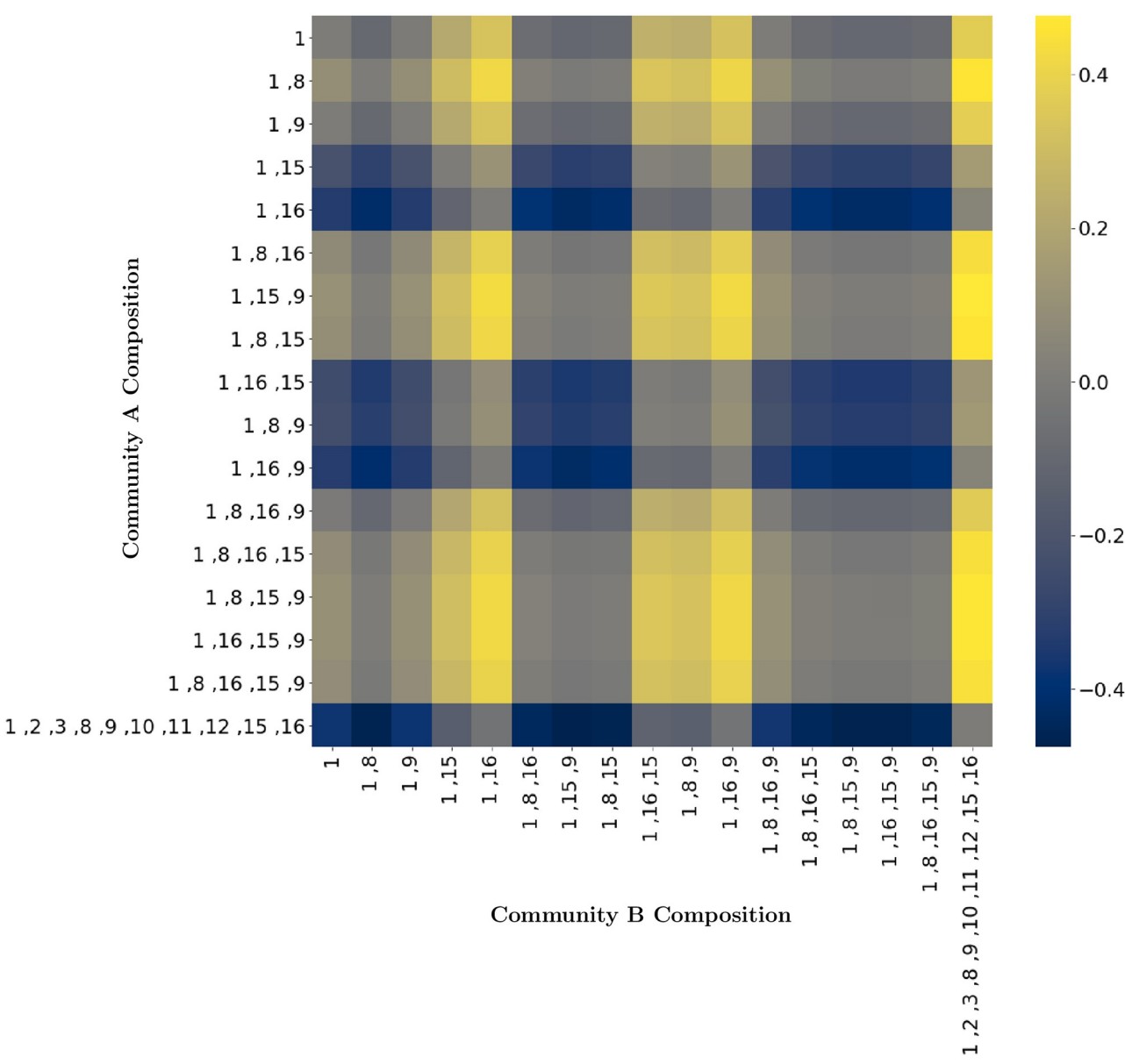

**Fig 6. The model of genome *bc1001* shows varying connections to the set of environmental metabolites when simulated in different isolated sub-communities.** Precisely, for each pair of communities shown involving *bc1001*, we computed the difference in strength (i.e. absolute value) of every connection that *bc1001* had in the time-averaged MetConSIN networks for both members of the pair, using absolute edge weight in community A minus absolute edge weight in community B. The heatmap displays the average of these differences across all shared connections for a pair.

abundance in the pair, as seen in Fig 7. These incorrect predictions highlight the fact that GSM reconstruction is often imperfect and focused on the core metabolism, while secondary metabolites often play a role in microbial interactions [52–54].

DFBA is a mechanistic model built from the underlying principles of metabolism and so its failure as a predictive model is not surprising. The main advantage of mechanistic models over better predicting (e.g. machine learning) models is their interpretability. MetConSIN provides this for DFBA with its ability to reveal the interactions implied by the GSMs provided. To demonstrate this advantage, we used MetConSIN to identify metabolite-mediated pathways

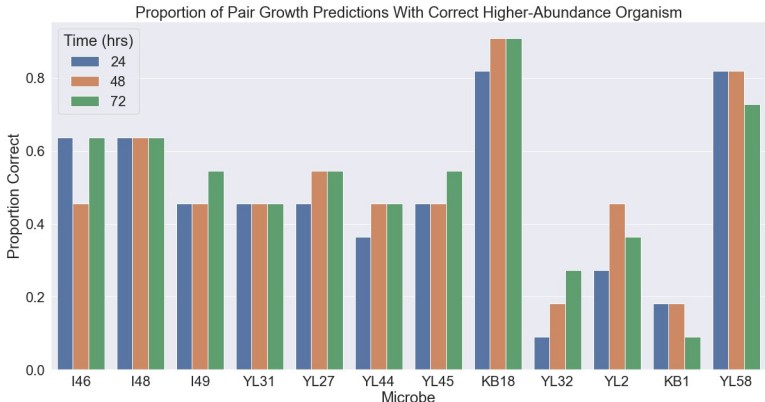

**Fig 7. DFBA simulations often failed to identify which of a pair would have higher abundance at any of the three experimental time-points.** For each microbe, we show the proportion of pair experiments involving that microbe for which DFBA correctly predicted the higher abundance member of the pair.

that matched (in sign) the observed directionality of the effective interactions implied by comparing pairwise growth with growth in monoculture. Following Weiss et al., we determined significant interactions using the t-test to compare an organism's growth in monoculture to its growth in a pair. For significantly different growth (p-value < 0.05) we determined directionality using the log-ratio of final time-point absolute abundance (note that Weiss et al. use the ratio, while we use the log-ratio simply so that values are mapped from $(0, \infty)$ to $(-\infty, \infty)$). For 32 of the 34 significant interactions, we identified metabolites that mediated interactions with the correct sign. In Fig 8, we show a heat-map of the the statistically significant interactions identified by Weiss et al. using pairwise growth experiments along with the strongest candidate mediating metabolite for that interaction as identified by MetConSIN.

## Comparisons with existing methods

To our knowledge, MetConSIN is the only method available for constructing the interaction networks implied by the DFBA mathematical model. It is possible to infer a rough approximation of the networks that MetConSIN provides by simply using the fluxes computed with FBA at a single time-point. However, this has several disadvantages compared to MetConSIN. First, FBA alone cannot accurately describe the effect of each metabolite on the growth rate of each microbe (i.e. arrows from metabolite to microbe). In fact, MetConSIN demonstrates that the concentrations of many metabolites that are consumed by a microbe may be perturbed with no effect on the microbial growth rate. A more accurate picture of the interactions happening within a community at specific point in time (or, equivalently, with a specific metabolic environment) could be produced by repeatedly computing FBA solutions while perturbing metabolites, but this would be extremely computationally expensive. In contrast, MetConSIN immediately provides a picture of which metabolites actually effect the growth of the microbes in a community, i.e. the rate-limiting metabolites. Second, MetConSIN provides information about interactions not just between metabolites and microbes, but also between metabolites and metabolites as mediated by microbes. In other words, MetConSIN shows how one metabolite can be effected by changing the availability of another. Finally, MetConSIN provides details about when, how and why a network changes qualitatively as a microbial community manipulates its environment, whereas a network based directly on flux balance can only provide details for a single discrete time-point. Therefore, discovering the transition points at

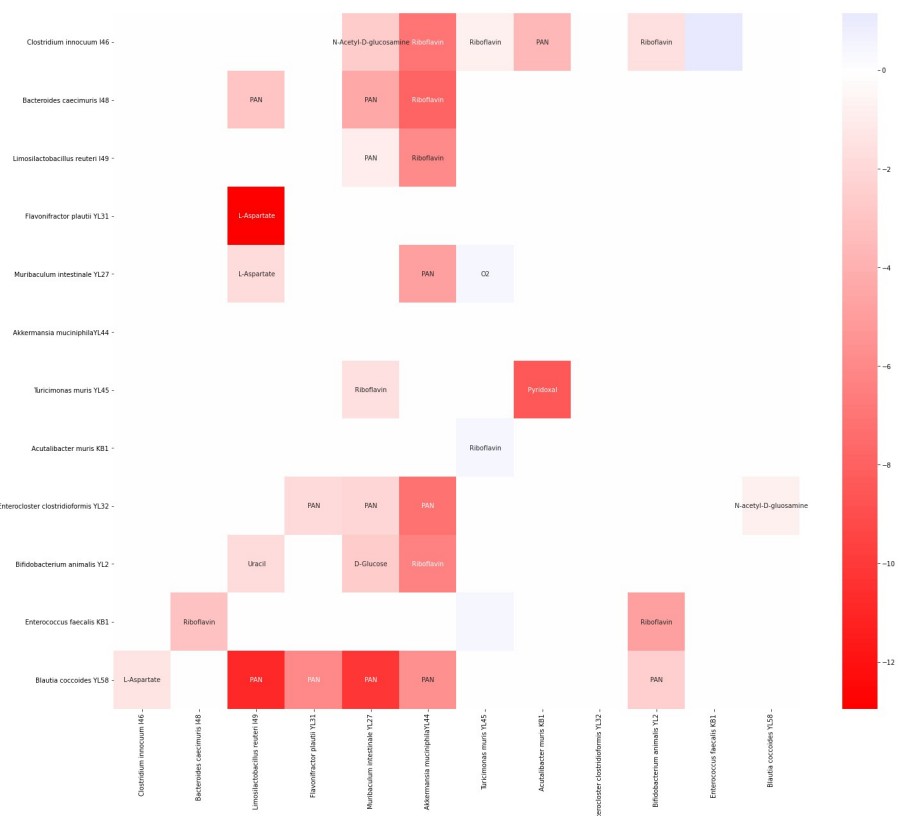

**Fig 8. MetConSIN is able to suggest candidate metabolites which mediate the significant interactions observed in growth experiments performed by Weiss et al.** Here, we show the statistically significant interactions among microbes as inferred by differences in monculture and paired growth along with a candidate metabolite that MetConSIN suggests might mediate that interaction.

which a community changes its behavior (i.e. basis changes in MetConSIN) would require relatively dense sampling with FBA.

To demonstrate the difference in networks inferred directly from FBA and networks constructed by MetConSIN, we created a method to infer a network from FBA by considering the fluxes of each reaction. These fluxes indicate how a microbe affects a metabolite, and so we added edges from microbe to metabolite weighted by the value of each flux. FBA provides no obvious way to assign edges from metabolites to microbes, so we assigned an edge from a metabolite to a microbe if the microbe consumed the metabolite. This reflects the assumption that a microbe will consume only the metabolites necessary for its growth. We note that such an edge should not be interpreted in the same way as a metabolite to microbe edge inferred by MetConSIN, which indicates that the metabolite in question directly affects the growth rate of the microbe (i.e. is rate limiting). We computed MetConSIN and direct FBA networks for a sample of small (2–4 member) communities chosen randomly from the 12 soil isolates. Fig 9 shows that there is some difference in weight of shared edges between MetConSIN networks and those constructed from FBA, and that networks constructed from FBA often contain many additional edges in comparison to MetConSIN networks.

As a result of the choice to consider every consumption of a metabolite indicated by FBA to give an edge from metabolite to microbe, networks inferred directly from FBA imply competition for many metabolites that are not rate limiting. These competitions in turn imply strong

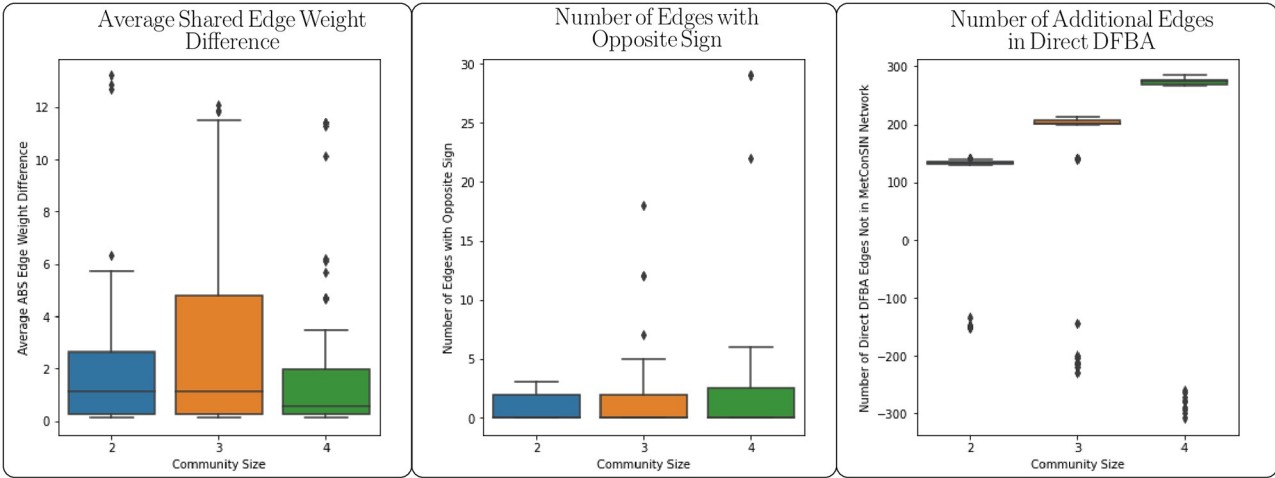

**Fig 9. Networks constructed directly from the solution to FBA at a time point in simulation show some difference in edge weight edges shared with MetConSIN networks constructed for the corresponding time interval, as well as a small number of edges with the opposite sign.** The major observable difference between the networks is that networks constructed directly from FBA contain a many edges that are spurious in the sense that the DFBA dynamical system (Eqs (3) and (4)) does not behave according the interactions represented by these edges. This is because FBA provides no direct information about how each environmental metabolite affects the growth rate of a microbe, and so some assumption must be made. In this case, we assumed that if microbe consumes a metabolite, then this metabolite positively affects the growth of that microbe.

negative interactions between microbes when this is not truely the case. Table 3 shows how MetConSIN identifies a set of species-species interactions based only on rate-limiting metabolites (note that in simulation oxygen was initially scarce), whereas Table 4 incorrectly identifies many additional interactions mediated by a host of metabolites that do not in fact limit growth of any microbe.

In Brunner & Chia [23], we demonstrated the theoretical improvement offered by *SurfinFBA*, the DFBA algorithm at the core of MetConSIN, by showing that *SurfinFBA* requires far fewer linear optimizations than a direct method of solving DFBA with a standard ODE solver, as well as a similar method that also makes use of forward simulation bases [22]. We used this metric of performance because it allows us to compare the theoretical algorithms behind available solvers, regardless of improvements gained by choice of language or code optimization.

In order to benchmark the time improvements provided by *SurfinFBA* and demonstrate that the solutions provided are similar to a direct repeated optimization approach, we implemented a direct solve method using *scipy*'s `solve_ivp` function [58]. This is fundamentally the same as the method suggested in the documentation of the CobraPy Toolbox [59]. We

**Table 3. MetConSIN species-species networks are based on shared interactions with the set of metabolites, and indicate which rate-limiting metabolites lead to an interaction.**

|  | Weight | Metabolites |
|---|---|---|
| bc1002→bc1008 | -0.032431 | D-Glucose |
| bc1012→bc1008 | -0.046521 | D-Glucose |
| bc1002→bc1012 | -0.152993 | O2 |
| bc1008→bc1012 | -0.003430 | O2 |
| bc1008→bc1002 | -0.003428 | O2 |
| bc1012→bc1002 | -0.152901 | O2 |

**Table 4. Species-species networks constructed from FBA cannot identify which metabolites lead to the interaction, and instead simply list all shared metabolites.** This also introduces the illusion of competition for many resources which are plentiful enough that competition does not occur, leading to strong negative edge weights.

|  | Weight | Metabolites |
|---|---|---|
| bc1002→bc1008 | -5570.901122 | 1,2-Diacyl-sn-glycerol dioctadecanoyl.4-Hydrox... |
| bc1012→bc1008 | -7717.811684 | 1,2-Diacyl-sn-glycerol dioctadecanoyl.4-Hydrox... |
| bc1002→bc1012 | -15620.771391 | 1,2-Diacyl-sn-glycerol dioctadecanoyl.4-Hydrox... |
| bc1008→bc1012 | -7717.811684 | 1,2-Diacyl-sn-glycerol dioctadecanoyl.4-Hydrox... |
| bc1008→bc1002 | -5570.901122 | 1,2-Diacyl-sn-glycerol dioctadecanoyl.4-Hydrox... |
| bc1012→bc1002 | -15620.771391 | 1,2-Diacyl-sn-glycerol dioctadecanoyl.4-Hydrox... |

simulated the DFBA problem using MetConSIN and the direct method for a sample of small (2–4 member) communities chosen randomly from the 12 soil isolates. In Fig 10, we show that *SurfinFBA* is indeed faster than the direct method and that solutions are very similar, with some slight error.

As an alternative to networks based directly on the solution to flux balance analysis, it is possible to construct a network of interactions between microbes by simulating knock-out or paired-growth experiments with community FBA approaches like *SteadyCom* [19] or *MICOM* [60]. However, this approach does not provide details about the interactions between microbes and metabolites, and assumes the community to be at equilibrium. Likewise, other DFBA tools, for example *COMETS* [61], can be used to infer some interactions by simulation of organisms in different combinations and with *in-silico* genetic modifications. In this way, *COMETS* was used to infer interactions within small 2 and 3 member communities [61]. However, this approach required repeated simulation to show that all community members needed to be present for growth and thus interactions were taking place, and inference of the interactions themselves was based on foreknowledge that allowed simulated gene knockout experiments. In contrast, MetConSIN, infers interactions from a single simulation without a need for further simulated experimentation to determine the details of the interactions.

Other DFBA tools are, with one notable exception, built on direct simulation by repeated optimization, using either Euler's method [62] or more sophisticated integration methods, as

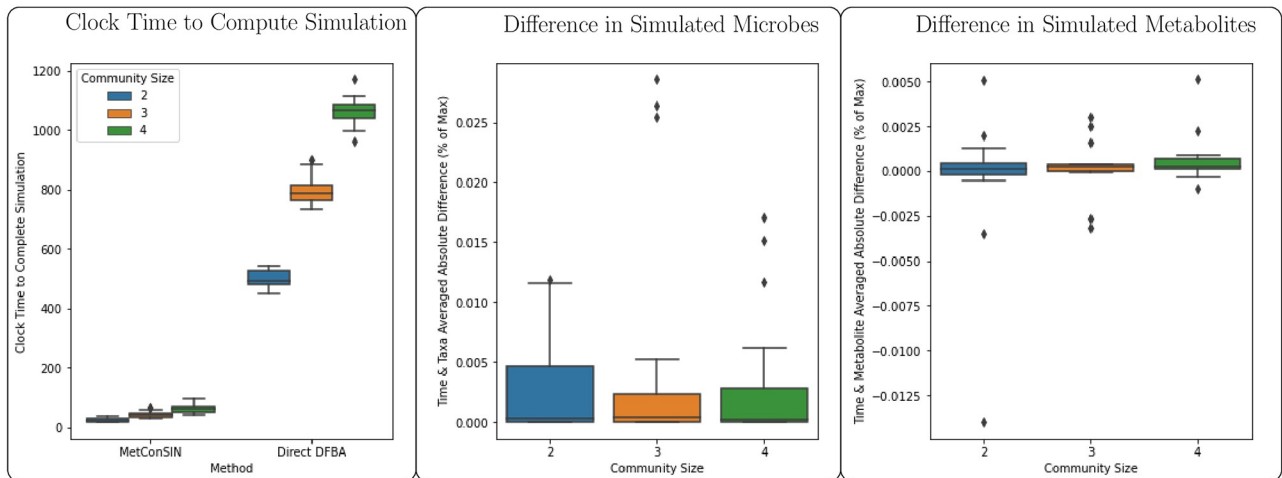

**Fig 10. Simulating with *SurfinFBA* (our simulation algorithm that is included in MetConSIN) was about 10 times faster than simulating using a direct method.** Solutions were generally very similar. Here, we present the time- and taxa-averaged absolute difference in simulation of microbes and time- and metabolite-averaged absolute difference in simulation of metabolites, relative to the max of the two simulations.

is suggested by the documentation of CobraPY [63]. For example, the popular *COMETS* tool [64] uses Euler's method with direct optimizations to simulate DFBA and integrates this into a spatial simulation. Likewise, *D-Optcom* [21] uses Euler's method to solve a dynamic problem related to DFBA which includes community-wide optimization. The exception to this rule is a pair of tools based on the method of Höffner et al. [65], a MatLab-based tool called *DFBALab* [66] and a python package called *dfba* [67]. The method described by Höffner is similar to *SurfinFBA*, although it differs in how bases are chosen for forward simulation. Unfortunately, *DFBALab* is currently not publicly available, and *dfba* does not to our knowledge allow for simulation of communities of more than one GEM.

## Limitations of DFBA & MetConSIN

DFBA provides a model for the population dynamics of microbial communities by leveraging genomic data. This means that dense time-longitudinal data is not required for simulation. Despite this important advantage, the usefulness of the DFBA model is limited. This is because a thorough qualitative analysis of the resulting dynamical simulation is often impractical due to the system's complexity. MetConSIN achieves an important step forward in analyzing DFBA simulations by organizing the complexity of DFBA into a sequence of interaction networks, which are more familiar and readily understood. This tool therefore gives researchers the power to infer important characteristics of the dynamic metabolic activity of a microbial community from genomic data.

MetConSIN depends on dynamic flux balance analysis and the genome-scale metabolic models that define that system. While this does mean that MetConSIN is essentially limited by the quality of the GSMs used, it also means that MetConSIN provides a method by which to assess the quality of these models. With high-quality GSMs, MetConSIN provides the ability to create qualitative predictions about community metabolic activity which can be used to generate testable hypotheses. MetConSIN can created testable hypotheses about the (1) resource competition and (2) community assembly dynamics in our synthetic communities. Furthermore, with MetConSIN, the accuracy of these hypotheses can be used to judge the usefulness of and ways to improve the underlying GSMs. Furthermore, MetConSIN provides a pathway for improving GSMs by comparing simulation to other "omics" data, e.g. metabolomics.

In the *in silico* experiments we performed with our 10 soil isolates, we lacked detailed information about an environment on which the interactions would take place. We assumed that every metabolite that the models could make use of was available, and calculated interactions based on this "uniform" media. For more precise study of known systems, the model can be constrained by information about the available environmental metabolites, e.g. using metabolomics data. We demonstrate this with our experiments using the data from Weiss et al. [55], in which the authors used metabolomic data to define a model environment. Alternatively, when modeling a well studied environment, environmental information can at times be found in the literature. In particular, the *Virtual Metabolic Human* project [68] define a number of environments corresponding to common human diets (e.g. "E.U. Average" or "High Fiber").

Higher-order interactions within microbial communities likely play an important role in community organization [69, 70]. However, the species-species interaction networks that we provide are fundamentally pairwise, and lack the capacity to convey the higher-order interactions that may be implied by the species-metabolite networks they are built from. In fact, any species-species model will lack the complexity necessary to reflect the possible interactions of a species-metabolite model [40, 41]. However, MetConSIN includes not just a single network, but a series of them. This means that it has the capacity to capture higher-order interactions

that emerge from the transitions between network structure. These represent changes in pairwise interactions caused by the actions of the community. Inferring higher order interactions or "effective" pairwise interactions that include higher-order effects [71] directly from the series of MetConSIN species-metabolite networks remains an ongoing area of research. We note also that higher-order effects are often mediated by secondary metabolites [54], which may be missing from the GSMs input into MetConSIN. Improving GSMs and automatic generation of GSMs remains an active area of research, and MetConSIN will benefit immediately from any improvements made in that area.

In addition to limitations of the underlying GSMs, interpretation of MetConSIN's results must also account for the inherent limitations of flux balance analysis. Whenever FBA is computed, it is possible that the optimal set of fluxes found is not unique. Like many methods based on FBA, MetConSIN attempts to mitigate this problem with a secondary optimization, by default minimizing the total flux through the metabolism of each taxa. Unlike other dynamic FBA methods, which must choose between non-unique solutions at every time-step, MetConSIN must only choose between optima at the very initial point of the simulation. However, MetConSIN will encounter non-uniqueness of the choice of basis for forward simulation at its discrete optimization (basis-change) time-points, meaning the resulting ODEs and networks are also non-unique. When this happens, the optimal solution found by MetConSIN is known as "degenerate", and it is exactly when the solution is degenerate that MetConSIN may change basis—without this degeneracy there would be no alternative basis to switch to. If only one possible basis allows forward simulation, MetConSIN will choose that basis and resulting networks. However, if the choice of basis is still non-unique even when constrained to allow forward simulation, MetConSIN chooses a basis by attempting to maximize the time until another basis is needed (details of this maximization can be found in supporting file S1 Text). This maximization is based on local (in time) information available to MetConSIN, reflecting the idea that a microbe will choose a strategy that will be more likely to work for a longer time based on the systems current state.

Ultimately, MetConSIN provides a rigorous interpretation of DFBA that emerges directly from the dynamics of the system. This tool is an important step in increasing the utility of genomic data and COBRA methods in the study of microbial communities and their impact on their environment.

## Supporting information

**S1 Table. Details of soil isolate sequencing experiments.**
(CSV)

**S1 Text. Technical details of *SurfinFBA*.**
(PDF)

## Acknowledgments

The authors would like to acknowledge the technical assistance of Thomas C. Biondi in this work.

## Author Contributions

**Conceptualization:** James D. Brunner, Marie E. Kroeger.

**Data curation:** James D. Brunner, Laverne A. Gallegos-Graves, Marie E. Kroeger.

**Formal analysis:** James D. Brunner.

**Funding acquisition:** Marie E. Kroeger.

**Investigation:** James D. Brunner, Laverne A. Gallegos-Graves, Marie E. Kroeger.

**Methodology:** James D. Brunner.

**Project administration:** Marie E. Kroeger.

**Resources:** Marie E. Kroeger.

**Software:** James D. Brunner.

**Supervision:** Marie E. Kroeger.

**Visualization:** James D. Brunner.

**Writing – original draft:** James D. Brunner.

**Writing – review & editing:** Marie E. Kroeger.

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
