## [Decision Letter · Decision Letter 0]

13 Sep 2023

Dear Dr. Brunner,

Thank you very much for submitting your manuscript "Inferring microbial interactions with their environment from genomic and metagenomic data" for consideration at PLOS Computational Biology.

As with all papers reviewed by the journal, your manuscript was reviewed by members of the editorial board and by several independent reviewers. In light of the reviews (below this email), we would like to invite the resubmission of a significantly-revised version that takes into account the reviewers' comments.

While all the major concerns raised by the reviewers need to be addressed, there are two major points that must be addressed thoroughly. The first is with respect to benchmarking, and importantly how this  method compares with existing methods, particularly the dynamic FBA models now widely used. While a 10-member community may be cumbersome to solve, a smaller community could be studied to establish the similarities and differences between the present method and the existing formalisms. Second, a comparison of how the interaction networks differ between their smooth simulations and numerical solvers would be critical to address.

We cannot make any decision about publication until we have seen the revised manuscript and your response to the reviewers' comments. Your revised manuscript is also likely to be sent to reviewers for further evaluation.

Sincerely,

Sunil Laxman, PhD

Academic Editor

PLOS Computational Biology

Stacey Finley

Section Editor

PLOS Computational Biology

Reviewer's Responses to Questions

**Comments to the Authors:**

Reviewer #1: Summary

The authors provide a tool – MetConSIN (Metabolically Contextualized Species Interaction Networks) - to efficiently simulate community metabolism in an interpretable way. Computational efficiency is achieved by reducing the number of optimizations performed to calculate the optimal flux in the metabolic network through the transformation of the linear programming problem to a set of differential equations, who’s solution can be carried forward in time without the need to re-optimize the parameters of the problem. This computational transformation also paves the way for the second, and main contribution of the new tool, which is interpretability. Because the transformation effectively produces an interaction network between organisms and metabolites, this allows for an immediate visual understanding of the causal interactions leading to metabolite, and ultimately species dynamics in the community.

Major comments

While I see no technical flaw in this work, I simply do not think it is innovative enough to warrant publication in a journal like PLoS Computational Biology. Many other tools exist that perform similar analysis of genome-scale metabolic models. In fact, excellent tools exists, such as COMETS (which is not cited here for some reason, and I am also not an author of), that are scalable (a major novelty claim in this paper) and even offer spatially resolved simulations. Specifically, COMETS automatically offers similar analysis, where the major difference is the automatic creation of interaction networks in MetConSIN, which is, in my opinion, not enough to warrant publication in a non-specialized journal. The authors also offer no comparison with existing methods (which was done in their previous publication) or with real world data, which make this paper quite “thin”. A comparison of how the interaction networks differ between their smooth simulations and numerical solvers would go a long way.

Finally, I think the figures with the networks would benefit from simplification. All of the nodes that are only connected by insignificant links (grey) could be removed, where the full network would be found in some supplementary figure. This would make reading the labels much easier and the interpretation and comparison of the different panels much easier, in my opinion.

Minor Comments

Line 146: Should it be “with non-decreasing c_ij^2 (y_j)”?

Line 184: non -> not

Line 340: Delete the word “in” in the sentence containing “but the strength of the competition may vary in with different”

Reviewer #2: The authors developed a mathematical approach that can extract microbe-metabolite interaction network from dynamic flux balance analysis (dFBA). This approach is based on their previous method to convert dFBA into piecewise ODEs. Various networks were constructed by interpreting the structure of ODEs. This is very interesting approach and I am very excited about it. I only have a few minor comments:

1. Each optimization step in dFBA is not unique in terms of which metabolites are consumed and produced. Are the equivalent ODEs unique or not? I am not familiar with the details of SurfinFBA; I guess the uniqueness maybe related to the choice of index set B. Since the reconstructed metabolic networks are derived from these ODEs, are they unique or not? It would be great if the authors add a comprehensive discussion about the uniqueness of dFBA, equivalent ODEs, and the reconstructed networks.

2. The rationale of developing this approach needs to be better explained. The optimal solution at each time point in dFBA depicts a community-level metabolic network: the exchanged metabolites of each GSM are known and their rates are quantitatively solved. Then why did the authors develop an indirect method to infer the network which requires conversion of dFBA to ODEs? I understand that ODEs are way easier to solve than dFBA but, in terms of network construction, how do you compare the pros and cons of the two approaches? And how different are between the metabolic networks inferred between the two approaches?

3. Both dFBA and ODEs are first-principle approaches. It would be super interesting if the approach can be extended to introduce constraints from metabolomics data. Could you discuss the possibility of integrating dFBA/ODEs with metabolomics data to infer realistic metabolic networks?

Reviewer #3: In this paper, the authors develop a nice method to perform dynamic FBA on multispecies microbial communities that is efficient and offers a time varying view of the interactions underlying the species and metabolites. Current dynamic FBA methods are time consuming and typically fail to offer insights into the interactions governing the dynamics. The present method overcomes both these limitations. The conceptual advance is in recognizing that the optimization problem governing FBA can be mapped to a system of linear equations and hence translated to a system of ODEs with parameters that remain constant over finite subintervals of time. The parameters change when metabolite concentrations change enough to violate the constraints on the original FBA problem. The authors present an elegant description of this new formalism and a software, MetConSIN, for implementing it. They apply it to a set of 10 soil microbial species, whose genomes they identify by sequencing and then construct genome scale metabolic models of each of the species to be used in MetConSIN. They deduce the interactions governing the species and the regimes over which the networks change.

The proposed method, in my opinion, represents a significant advance over existing methods because of its ability to offer time varying interaction maps and hence insights not readily gained by existing methods. The computational gains are a bonus.

The paper is well written overall. I have a few comments for the authors to consider.

Major comments:

1. My first comment is with respect to benchmarking. While the authors demonstrate the applicability of their method to the 10-member soil community, they do not show how their method compares with existing methods, particularly the dynamic FBA set up in Eqs. (1)-(4). Are the time courses predicted in Fig. 2 similar to what might be expected from Eqs. (1)-(4)? If the 10-member community is cumbersome to solve, can a smaller subcommunity – even a 2 member community – be studied to establish the similarities and differences between the present method and the existing formalisms?

2. Along the same lines, is a comparison with any experimental system feasible? The authors seem to have cultured the 10-members they studied. Could their growth rates be monitored in multi-species cultures and then compared with corresponding model predictions? If co-culturing is not possible, are other previously published datasets amenable to comparisons with the present model? If this not possible too, the authors must discuss this and mention explicitly what prevents comparisons with experiments. The difficulty may exist with current methods too, in which case, this may not be a limitation of the present study alone, but it must be discussed nonetheless.

3. The analysis of the interaction networks and their evolution (Figs 2-4) is very nice. It highlights the strength of the method. I felt though that the interpretation of the transitions seen seemed somewhat superficial. The authors mention that the first transition is when bc1012 altered its connectivity three times in quick succession (lines 286-288). They, however, do not provide any explanation of these changes in connectivity. Thus, while knowledge of these transitions is indeed an advance over existing models and is thus welcome, a mechanistic understanding of the transitions based on the metabolic models of the species and the nutrients available could have been more satisfying. Are these explanations forthcoming? If not, the authors must discuss why.

4. My final comment is on the way interactions between species are deduced (Eq. 13). Pairs of species are chosen and their interactions mediated by metabolites are summed with suitable weights to yield the net interactions between the species. This method yields pairwise interactions. However, species often experience high-order interactions (e.g., see: 1) https://www.pnas.org/doi/10.1073/pnas.1809349115; 2) https://www.nature.com/articles/s43588-021-00131-x). Does the present method thus miss these high-order interactions? Because the dynamic FBA formalism does not make any assumptions on the interactions but only deduces them, any high-order interactions present must exist in the model calculations. The deduction method may have to be changed to consider triplets of species, quadruplets of species, etc. (instead of just pairs) in order to deduce third-order, fourth-order, etc. interactions. I wonder if currently, the pairwise interactions deduced yield ‘effective’ pairwise interactions, as has been suggested in the recent study above (https://www.nature.com/articles/s43588-021-00131-x)? Again, I feel that the authors must at least comment on high-order interactions, given their possible presence in multi-species communities and the focus of the present study on deducing interaction networks.

Minor comment:

1. On lines 126-131, the authors indicate that the method to choose the matrices (Bik) are outlined elsewhere. For completeness, I feel that the authors may wish to provide a brief outline of how this choice is made.

2. On lines 140-142, the authors mention that holding the metabolite levels constant would yield a snapshot of the interaction may between the species. Could this be shown? Also, I would imagine that the species compositions would evolve with time even if the metabolite levels were held constant. Then, would the interaction map not also change? The authors may wish to comment on this.

**Have the authors made all data and (if applicable) computational code underlying the findings in their manuscript fully available?**

Reviewer #1: Yes

Reviewer #2: Yes

Reviewer #3: Yes

PLOS authors have the option to publish the peer review history of their article (what does this mean?). If published, this will include your full peer review and any attached files.

Reviewer #1: No

Reviewer #2: No

Reviewer #3: No
---

## [Decision Letter · Decision Letter 1]

4 Nov 2023

Dear Dr. Brunner,

We are pleased to inform you that your manuscript 'Inferring microbial interactions with their environment from genomic and metagenomic data' has been provisionally accepted for publication in PLOS Computational Biology.

Best regards,

Sunil Laxman, PhD

Academic Editor

PLOS Computational Biology

Stacey Finley

Section Editor

PLOS Computational Biology

Reviewer's Responses to Questions

**Comments to the Authors:**

Reviewer #1: The authors addressed all of my concerns and I have no further comments.

Reviewer #3: I am impressed with the work that the authors have done to address my concerns. I am quite satisfied with their responses and have no further comments.

**Have the authors made all data and (if applicable) computational code underlying the findings in their manuscript fully available?**

Reviewer #1: Yes

Reviewer #3: Yes

PLOS authors have the option to publish the peer review history of their article (what does this mean?). If published, this will include your full peer review and any attached files.

Reviewer #1: No

Reviewer #3: No

---

## [Editor Report · Acceptance letter]

9 Nov 2023

PCOMPBIOL-D-23-01066R1 

Inferring microbial interactions with their environment from genomic and metagenomic data

Dear Dr Brunner,

I am pleased to inform you that your manuscript has been formally accepted for publication in PLOS Computational Biology. Your manuscript is now with our production department and you will be notified of the publication date in due course.

With kind regards,

Anita Estes
